

# Multistable Slip of a One-degree-of-freedom Spring-slider Model in the Presence of Thermal-pressurized Slip-weakening Friction and Viscosity

Jeen-Hwa Wang

Institute of Earth Sciences, Academia Sinica

P.O. Box 1-55, Nangang, Taipei, TAIWAN

e-mail: jhwang@earth.sinica.edu.tw

**Abstract** This study is focused on multistable slip of earthquakes based on a one-degree-of-freedom slider-slider model in the presence of thermal-pressurized slip-weakening friction and viscosity by using the normalized equation of motion of the model. The major model parameters are the normalized characteristic displacement, $U_c$, of the friction law and the normalized viscosity coeficient, $\eta$, between the slider and background plate. Analytic results at small slip suggest that there is a solution regime for $\eta$ and $\gamma$ ($=1/U_c$) to make the slider slip steadily. Numerical simulations exhibit that the time variation in normalized velocity, $V/V_{max}$ ($V_{max}$ is the maximum velocity), obviously depends on $U_c$ and $\eta$. The effect on the amplitude is stronger due to $\eta$ than due to $U_c$. In the phase portrait of $V/V_{max}$ versus the normalized displacement, $U/U_{max}$ ($U_{max}$ is the maximum displacement), there are two fixed points. The one at large $V/V_{max}$ and large $U/U_{max}$ is not an attractor; while that at small $V/V_{max}$ and small $U/U_{max}$ can be an attractor for some values of $\eta$ and $U_c$. When $U_c<0.55$, unstable slip does not exist. When $U_c\geq0.55$, $U_c$ and $\eta$ divide the solution domain into three regimes: stable, intermittent, and unstable (or chaotic) regimes. For a certain $U_c$, the three regimes are controlled by a lower bound, $\eta_l$, and an upper bound, $\eta_u$, of $\eta$. The values of $\eta_l$, $\eta_u$, and $\eta_u-\eta_l$ all decrease with increasing $U_c$, thus suggesting that it is easier to yield unstable slip for larger $U_c$ than for smaller $U_c$ or for larger $\eta$ than for smaller $\eta$. When $U_c<1$, the Fourier spectra calculated from simulation velocity waveforms exhibit several peaks, thus suggesting the existence of nonlinear behavior of the system. When $U_c>1$, the related Fourier spectra show only



one peak, thus suggesting linear behavior of the system.
**Key Words**: Multistable slip, one-degree-of-freedom spring-slider model,
displacement, velocity, thermal-pressurized slip-weakening friction, viscosity

**1. Introduction**

The earthquake ruptures consist of three steps: nucleation, dynamical

propagation, and arrest. Due to the lack of a comprehensive model, a set of equations
to completely describe fault dynamics has not yet been established, because
earthquake ruptures are very complicated. Nevertheless, some models, for instance
the crack model and dynamical lattice model, have been developed to approach fault
dynamics. Several factors will control earthquake ruptures (see Wang, 2016b; and
cited references herein), including at least brittle-ductile fracture rheology, normal
stress, re-distribution of stresses after fracture, fault geometry, friction, seismic
coupling, pore fluid pressure, elastohydromechanic lubrication, thermal effect,
thermal pressurization, and metamorphic dehydration. A general review can be seen in
Bizzarri (2009). Among the factors, friction and viscosity are two important ones in
controlling faulting.

Burridge and Knopoff (1967) proposed a one-dimensional spring-slider model

(abbreviated as the 1-D BK model henceforth) to approach fault dynamics. Wang
(2000, 2012) extended this model to a two-dimensional version. The two models and
their modified versions have been long and widely applied to simulate the occurrences
of earthquakes (see Wang, 2008, 2012; and cited references therein). In the followings,
the one-, two-, three-, few-, and many-body models are used to represent the one-,
two-, three-, few-, and many-degree-of-freedom spring-slider models, respectively.
The few-body models have been long and widely used to approach faults (Turcotte,

1992)

Since the commonly-used friction laws are nonlinear, the dynamical model itself

could behave nonlinearly. A nonlinear dynamical system can exhibit chaotic
behaviour under some conditions (Thompson and Stewart, 1986; Turcotte, 1992).
This means that the system is highly sensitive to initial conditions (SIC) and thus a
small difference in initial conditions, including those caused by rounding errors in
numerical computation, yields widely diverging outcomes. This indicates that
long-term prediction is impossible in general, even though the system is deterministic,



meaning that its future behavior is fully determined by their initial conditions, without
random elements. This behavior is known as (deterministic) chaos (Lorenz, 1963).
An interesting question is: Can a simple few-body model with total symmetry
make significant predictions for fault behavior? Gu et al. (1984) first found some
chaotically bounded oscillations based on a one-body model with rate- and state-
dependent friction. Perez Pascual and Lomnitz-Adler (1988) studied the chaotic
motions of coupled relaxation oscillators. Related studies have been made based on
different spring-slider models: (1) a one-body model with rate- and state-dependent
friction (e.g., Gu et al., 1984; Belardinelli and Belardinelli, 1996; Ryabov and Ito,
2001; Erickson et al., 2008, 2011; Kostić et al., 2013); (2) a one-body model with
velocity-weakening friction (e.g., Brun and Gomez, 1994); (3) a one-body model with
slip-weakening friction (e.g., Wang, 2016a,b); (4) a two-slider model with simple
static/dynamic friction (e.g., Nussbaum and Ruina, 1987; Huang and Turcotte, 1990);
(5) a two-body model with velocity-dependent friction (e.g., Huang and Turcotte,
1992; de Sousa Vieira, 1999; Galvanetto, 2002); (6) a two-body model with rate- and
state-dependent friction (e.g., Abe and Kato, 2013); (7) a two-body model with
velocity-weakening friction (Brun and Gomez, 1994); (8) a two-body model with
slip-weakening friction (e.g., Wang, 2017); (9) many-body model with velocity-
weakening friction (e.g., Carlson and Langer, 1989; Wang, 1995, 1996); and (10)
one-body quasi-static model with rate- and state-dependent friction (e.g., Shkoller and
Minster, 1997). Results suggest that predictions for fault behaviour are questionable
due to the possible presence of chaotic slip.
The frictional effect on earthquake ruptures has been widely studied as
mentioned above. However, the studies of viscous effect on earthquake ruptures are
rare. The viscous effect mentioned in Rice et al. (2001) was just an implicit factor
which is included in the evolution effect of friction law. In this work, I will investigate
the effects of thermal pressurized slip-weakening friction and viscosity on earthquake
ruptures and the generation of unstable (or chaotic) slip based on a one-body model.

**2. MODEL**
**2.1 One-body Model**
Fig. 1 shows the one-body model whose equation of motion is:

$$m d^2u/dt^2 = -K(u-u_o) - F(u,v) - \Phi(v),$$    (1)




where m is the mass of the slider, u and v (=du/dt) are, respectively, the displacement
and velocity of the slider, $u_o$ is the equilibrium location of the slider, K is the spring
constant, F is the frictional force between the slider and the background and a
function of u or v, and $\Phi$ is the viscous force between the slider and the background
and a function of v. The slider is pulled by a driving force $F_D$ due to the moving plate
with a constant driving velocity, $v_p$, through a leaf spring of strength, K. Hence, the
driving force is $F_D=Kv_pt$ and thus $u_o=v_pt$. When $F_D$ is slightly larger than the static
frictional force, $F_o$, friction changes from static friction strength to dynamic one and
thus the slider moves.

**2.2 Viscosity**

Jeffreys (1942) first emphasized the importance of viscosity on faulting.
Frictional melts in faults depend on temperature, pressure, water content, and etc.
(Turcotte and Schubert, 1982) and can yield viscosity on the fault plane (Byerlee,
1968). Rice et al. (2001) discussed that rate- and state-dependent friction in thermally
activated processes allows creep slippage at asperity contacts on the fault plane.
Scholz (1990) suggested that the friction melts would present significant viscous
resistance to shear and thus inhibit continued slip. However, Spray (1993, 1995, 2005)
stressed that the frictional melts possessing low viscosity could generate a sufficient
melt volume to reduce the effective normal stress and thus act as fault lubricants
during co-seismic slip. His results show that viscosity remarkably decreases with
increasing temperature. For example, Wang (2011) assumed that quartz plasticity
could be formed in the fault zone when T>300 $^o$C after faulting and it would lubricate
the fault plane at higher T and yield viscous stresses to resist slip at lower T. From
numerical simulations, Wang (2007, 2016b, 2017) stressed the viscous effect on
faulting. Noted that several researchers (Knopoff et al., 1973; Cohen, 1979; Xu and
Knopoff, 1994; Knopoff and Ni, 2001; Dragoni and Santini, 2015) took viscosity as a
factor in causing seismic radiation to reduce energy during faulting.
The viscosity coefficient, $\upsilon$, of rocks is mainly controlled by temperature, T. An
increase in T will yield partial melting of rocks and thus the viscosity coefficient, $\upsilon$,
first is increased, then reaches the largest value at a particular T, and finally decreases
with increasing T The relation between $\upsilon$ and T can be described by the following
equation (e.g., Turcotte and Schubert, 1982): $\upsilon=\upsilon_o\exp[(E_o+pV_a/RT)]$ where $\upsilon_o$ is the





largest viscosity at low ambient T of an area, $E_o$ is the activation energy per mole, p is
the pressure, $V_a$ is the activation volume per mole, and R is the universal gas constant
($E_o/R \approx 3 \times 10^4$ K). Obviously, $\upsilon$ decreases with increasing T. This is particularly
remarkable in regions of high confining pressure. On the other hand, Diniega et al.
(2013) assume that $\upsilon$ exponentially depends on temperature: $\upsilon \sim e^{\beta(1-T^*)}$, where $\beta$ is a
constant and $T^* = (T-T_C)/(T_H-T_C)$ is a dimensionless temperature within a
temperature range of $T_C$ to $T_H$. The value of $\upsilon$ increases with $T^*$ when $T^*<1$ and
decreases with increasing $T^*$ when $T^*>1$. Wang (2011) inferred that in the major slip
zone<0.01 m of the 1999 $M_s$7.6 Chi-Chi, Taiwan, earthquake, T(t) in the fault zone at
a depth of 1111 m increased from ambient temperature $T_a \approx 45$ $^o$C at t=0 s to peak
temperature $T_{peak}$=1135.1 $^o$C at t=~2.5 s. T(t) began to decrease after t=2.5 s and
dropped to 160 $^o$C at t=195 s. This yields a change of viscosity in the fault zone.

The description about the physical models of viscosity can be found in several

articles (Jaeger and Cook, 1977; Cohen, 1979; Hudson, 1980; Wang, 2016b). A brief
description is given below. For many deformed materials, there are elastic and viscous
components. The viscous component can be modeled as a dashpot such that the
stress–strain rate relationship is: $\sigma=\upsilon(d\varepsilon/dt)$ where $\sigma$ and $\varepsilon$ are the stress and the strain,
respectively. Two simple models (shown in Fig. 2) commonly used to describe the
viscous materials are the Maxwell model and the Kelvin-Voigt model (or the Voigt
model). The first one can be represented by a purely viscous damper (denoted by "D")
and a purely elastic spring (denoted by "S") connected in series,. Its constitution
equation is: $d\varepsilon/dt=d\varepsilon_D/dt+d\varepsilon_S/dt=\sigma/\upsilon+E^{-1}d\sigma/dt$ where E is the elastic modulus and
$\sigma=E\varepsilon$. The constitutive relation of the second model is: $\sigma(t)=E\varepsilon(t)+\upsilon d\varepsilon(t)/dt$.

For the Maxwell model, the strain will increase, without a upper limit, with time;

while the Kelvin-Voigt model the strain will increases, with a upper limit, with time.
Wang (2016b) assumed that the latter is more appropriate than the former to be
applied to the seismological problems as suggested by Hudson (1980). Hence, the
Kelvin-Voigt model is taken in this study. To simplify the problem, only a constant
viscosity is considered below. The viscous stress at the slider is represented by -$\upsilon$v.

However, it is not easy to directly implement viscosity in a dynamical system as

used in this study. Wang (2016b) represented the viscosity coefficient in an alternative
way. Viscosity leads to the damping of oscillations of a body in viscous fluids. The
damping coefficient, $\eta$, depends on the viscosity coefficient, $\upsilon$, and the linear


dimension, R, of the body in a viscous fluid. According to Stokes' law, the $\eta$ of a
sphere of radius R in a viscous fluid of $\upsilon$ is $\eta=6\pi R\upsilon$ (cf. Kittel et al., 1968). In order
to simplify the problem, the damping coefficient is taken in this study. Hence, the
viscous force is $\Phi=\eta v$. Noted that the unit of $\eta$ is $N(m/s)^{-1}$.

**2.3 Friction caused by thermal pressurization**

Numerous factors can affect friction (see Wang, 2009, 2016b; and cited
references herein). When fluids are present and temperature changes in faults, thermal
pressurization will yield resistance on the fault plane and thus play a significant role
on earthquake rupture (Sibson, 1973; Lachenbruch, 1980; Chester and Higgs, 1992;
Fialko, 2004; Fialko and Khzan, 2005; Bizzari and Cocco, 2006a,b; Rice, 2006; Wang,
2000, 2006, 2009, 2011, 2013, 2016b, 017; Bizzarri, 2010; Bizzarri, 2011a,b).
Rice (2006) proposed two end-members models for thermal pressurization: the
adiabatic-undrained-deformation (AUD) model and slip-on-a-plane (SOP) model. He
also obtained the shear stress-slip functions caused by the two models. The first model
corresponds to a homogeneous simple shear strain $\varepsilon$ at a constant normal stress $\sigma_n$ on
a spatial scale of the sheared layer that is broad enough to effectively preclude heat or
fluid transfer. The second model shows that all sliding is on the plane with $\tau(0)=$
$f(\sigma_n-p_o)$ where $p_o$ is the pore fluid pressure on the sliding plane (y=0). For this second
model, heat is transferred outwards from the fault plane. Although the stress $\tau_{sop}(u)$
also shows slip-weakening (Wang, 2009), the SOP model is not appropriate in this
study because of the request of a constant velocity for this model.
The shear stress-slip functions, $\tau(u)$, caused by the AUD model is:
$\tau_{aud}(u)= f(\sigma_n-p_o)\exp(-u/u_c).$                (3)
The parameters $u_c$ is the characteristic displacements associated with the thickness
and some physical properties of fault zone. The stress $\tau_{aud}(u)$ displays exponentially
with u and thus exhibits slip-weakening friction. Based on the AUD model, Wang
(2009) proposed a simplified slip-weakening friction law (denoted by the TP law
hereafter): $F(u)=F_o\exp(-u/u_c)$, where $F_o$ is the static frictional force, to study seismic
efficiency. Wang (2016b, 2017) applied the law to simulate slip of one-body and
two-body spring-slider models. Fig. 3 exhibits F(u) versus u for five values of $u_c$, i.e.,
0.1, 0.3, 0.5, 0.7, and 0.9 m. The friction force decreases with increasing u and it





decreases faster for smaller $u_c$ than for larger $u_c$. Meanwhile, the force drop decreases
with increasing $u_c$. For small u, $\exp(-u/u_c)$ can be approximated by $1-u/u_c$ (Wang,
2016a,b, 2017). The parameter $u_c^{-1}$ is almost the decreasing rate, $\gamma$, of friction force
with slip at small u. Small (large) $u_c$ is related to large (small) $\gamma$.
**2.4. Predominant Frequency and Period of the System**
To conduct marginal analyses of slip of one-body model with friction, Wang
(2016b) used the friction law: $F(u)=F_o-\gamma u$. His results show that the natural periods
are $T_o=2\pi/(K/m)^{1/2}$ when friction and viscosity are excluded and

$T_n=T_o/[1-T_o^2(\eta^2+4m\gamma)/(4\pi m)^2]^{1/2}.$    (4)

when friction and viscosity are included. Clearly, $T_n$ is longer than $T_o$. Eq. (4) shows
that $T_n$ increases with $\eta$ and $\gamma$, thus indicating that friction and viscosity both lengthen
the natural period of the system.

**3. Normalization of Equation of Motion**
Substituting the TP law and the linear viscous law into Eq. (1) leads to

$md^2u/dt^2=-K(u-u_o)-F_o\exp(-u/u_c)-\eta v.$    (5)

To simplify numerical computations, Eq. (5) is normalized based on the following
normalization parameters: $D_o=F_o/K$, $\omega_o=(K/m)^{1/2}$, $\tau=\omega_o t$, $U=u/D_o$, $U_c=u_c/D_o$, and
$\Gamma_D=F_D/K$. This gives $du/dt=[F_o/(mK)^{1/2}] dU/d\tau$, $d^2u/dt^2=(F_o/mK)d^2U/d\tau^2$. The driving
velocity becomes $V_p=v_p/D_o\omega_o$ Hence, the normalized acceleration and velocity are,
respectively, $A=d^2U/d\tau^2$ and $V=dU/d\tau$. The phase $\omega t$ is replaced by $\Omega\tau$, where
$\Omega=\omega/\omega_o$ is the dimensionless angular frequency. Note that $\eta/(mK)^{1/2}$ is simply
denoted by $\eta$ below. Clearly, all normalization parameters are dimensionless. Hence,
Eq. (5) becomes:

$d^2U/d\tau^2=-U-\eta dU/d\tau-\exp(-U/U_c)+\Gamma_D.$    (6)

When $F_D=v_p t$ or $\Gamma_D=V_p\tau$, Eq. (6) is transformed to a set of three first-order





differential equations by defining $x=U/U_c$, $y=V/V_p$, and $z=-U+V_p\tau-\eta V_p y_\tau$
($y_t=dy/d\tau$):

$x_\tau=(V_p/U_c)y$                   (7a)


$y_\tau=(z-e^{-x})/V_p,$              (7b)


$z_\tau=V_p(1-y-\eta y_\tau).$         (7c)


As $x\ll1$, $e^{-x}\approx1-x$ and thus Eq. (7b) can be approximated by $y_\tau\approx(z-1+x)/V_p$. The

condition of $x\ll1$ shows $U/U_c\ll1$. Differential of this equation leas to
$y_{\tau\tau}\approx(z_\tau+x_\tau)/V_p$, where $y_{\tau\tau}=d^2y/d\tau^2$. Substituting Eqs. (7a) and (7b) into this equation
gives

$y_{\tau\tau}+\eta y_\tau+(1-1/U_c)y=1.$        (8)


The homogeneous equation of Eq. (8) is

$y_{\tau\tau}+\eta y_\tau+(1-1/U_c)y=0.$        (9)


Let the general solution be $y\sim e^{\lambda\tau}$. This leads to $[\lambda^2+\eta\lambda+(1-/U_c)]y=0$ or

$\lambda^2+\eta\lambda+(1-/U_c)=0.$          (10)


The solutions of Eq. (10) are

$\lambda_\pm=-\eta/2\pm[\eta^2-4(1-1/U_c)]^{1/2}/2.$      (11)


The term $-\eta/2$ of Eq. (11) leads to $e^{-\lambda/2}$ which yields attenuation of y. Define $D(\eta,1/U_c)$
to be $\eta^2-4(1-1/U_c)$. As mentioned above, $U_c^{-1}$ is the normalized decreasing rate of
friction, $\Gamma$, at $U=0$. Fig. 4 shows the plot of $\eta$ versus $1/U_c$ and thus exhibits the root
structure of the system. Because $\eta>0$ and $U_c>0$, only the plot in the first quadrant is





present in Fig. 4. The solid line displays the function: $D(\eta, 1/U_c) = \eta^2 - 4(1 - 1/U_c) = 0$.
Along the line, we have $\eta^2 = 4(1 - 1/U_c)$, and thus $\lambda_\pm = -\eta/2$. In other word, the roots are
equal and real, and thus the solution is a stable inflected node displayed by a solid
circle in Fig. 4. As $D(\eta, 1/U_c) > 0$ or $\eta^2 > 4(1 - 1/U_c)$, the roots are both real and negative.
The solution shows no oscillation and thus is a stable node shown by a solid square in
Fig. 4. As $D(\eta, 1/U_c) < 0$ or $\eta^2 < 4(1 - 1/U_c)$, the roots are complex with negative real part.
This results in oscillations of exponentially decaying amplitude. The solution is a
stable spiral or a stable focus shown by an open circle in Fig. 4.

**4. Numerical Simulations**
Let $y_1 = U$ and thus $y_2 = dU/d\tau$. Eq. (6) can be re-written as two first-order
differential equations:

$$dy_1/d\tau = y_2 \tag{12a}$$

$$dy_2/d\tau = -y_1 - \eta y_2 - \exp(-y_1/U_c) + \Gamma_D. \tag{12b}$$

Eq. (12) will be numerically solved using the fourth-order Runge-Kutta method (Press
et al., 1986). To simplify the following computations, the value of $\Gamma_D$ is set to be a
small constant of $10^{-5}$, which can continuously enforce the slider to move.
A phase portrait, denoted by $y = f(x)$, is a plot of a physical quantity versus
another of an object in a dynamical system (Thompson and Stewart, 1986). The
intersection point of the bisection line, i.e., $y = x$, and $f(x)$ is called the fixed point, that
is, $f(x) = x$. If the function $f(x)$ is continuously differentiable in an open domain near a
fixed point $x_f$ and $|f'(x_f)| < 1$, attraction is generated. In other words, an attractive fixed
point is a fixed point $x_f$ of a function $f(x)$ such that for any value of $x$ in the domain
that is close enough to $x_f$, the iterated function sequences, i.e., $x$, $f(x)$, $f^2(x)$, $f^3(x)$,…,
converges to $x_f$. An attractive fixed point is a special case of a wider mathematical
concept of attractors. Chaos can be generated at some attractors. The details can be
seen in Thompson and Stewart (1986) or other nonlinear literatures. In the following
plots, the intersection points of the bisection line (denoted by a thin solid line) with
the phase portrait of $V/V_{max}$ versus $U/U_{max}$ are the fixed points. To explore nonlinear
behavior of a system, the Fourier spectrum $F[V(\Omega_k)]$, where $\Omega_k = k/\delta\tau$ is the





297 dimensionless angular frequency at k=0, ..., N-1, is calculated for the simulation

298 velocity waveform through the fast Fourier transform (Press et al., 1986). The

299 bifurcation from a predominant period to others will be seen in the Fourier spectra.

300  Numerical simulations for the time variation in $V/V_{max}$, the phase portrait of

301 $V/V_{max}$ versus $U/U_{max}$, and Fourier spectrum based on different values of model

302 parameters are displayed in Figs. 5–12. In the figures, $V_{max}$ and $U_{max}$ are, respectively,

303 the maximum velocity and displacement for case (a) of each figure, because the

304 maximum values of U and V decrease from case (a) to case (d) in this study.

305  First, the cases excluding viscosity, i.e., η=0, are explored. Fig. 5 is numerically

306 made for four values of $U_c$: (a) for $U_c$=0.1; (b) for $U_c$=0.4; (c) for $U_c$=0.7; and (d) for

307 $U_c$=0.9 when η=0. Fig. 6 is numerically made for four values of $U_c$: (a) for $U_c$=1.00;

308 (b) for $U_c$=1.01; (c) for $U_c$=1.15; and (d) for $U_c$=2.00 when η=0. A comparison

309 between Fig. 5 and Fig. 6 suggests that $U_c$=1 is a transition value of the friction law

310 between two modes of slip as displayed in Fig. 4. Only $U_c$<1 is considered below.

311  Secondly, the cases including both friction and viscosity are studied. Fig. 7 is

312 numerically made for four values of η: (a) for η=0.20; (b) for η=0.50; (c) for η=0.87;

313 and (d) for η=0.90 when $U_c$=0.20. Obviously, the time variation in $V/V_{max}$ exhibits

314 cyclic oscillations with a particular period when η<$η_l$=0.86 and has intermittent slip

315 with shorter periods when η>$η_l$. Such a phenomenon holds also for η<5.5.

316  Fig. 8 is numerically made for four values of η: (a) for η=0.46; (b) for η=0.47; (c)

317 for η=0.98; and (d) for η=0.99 when $U_c$=0.55. The Fourier spectrum is not calculated

318 for case (d), because the velocity becomes negative infinity at a certain time. The time

319 variation in $V/V_{max}$ exhibits cyclic oscillations specified with three main frequencies

320 when η<$η_l$=0.47. There is intermittency slip with shorter periods when

321 $η_l$<η<$η_u$=0.98. There are unstable slip when η>$η_u$. This phenomenon holds also

322 when 0.55<$U_c$<1.

323  Four examples for η varying from η<$η_u$ to η>$η_u$ for different values of $U_c$ are

324 displayed in Figs. 9–12. Fig. 9 is made for four values of η: (a) for η=0.39; (b) for

325 η=0.83; (c) for η=0.84; and (d) for η=0.85 when $U_c$=0.6. Fig. 10 is made for four

326 values of η: (a) for η=0.34; (b) for η=0.71; (c) for η=0.72; and (d) for η=0.73 when

327 $U_c$=0.7. Fig. 11 is made for four values of η: (a) for η=0.25; (b) for η=0.53; (c) for

328 η=0.54; and (d) for η=0.55 when $U_c$=0.8. Fig. 12 is made for four values of η: (a) for





$\eta=0.14$; (b) for $\eta=0.35$; (c) for $\eta=0.36$; and (d) for $\eta=0.37$ when $U_c=0.9$. The Fourier
spectrum is not calculated for case (d) in each example, because the velocity becomes
negative infinity at a certain time.
Fig. 13 exhibits the data points of $\eta_l$ (with a solid square) and that of $\eta_u$ (with a
solid circle) for several values $U_c$. The values of $\eta_l$ and $\eta_u$ for several values of $U_c$
are given in Table 1. The figure exhibits a stable regime when $\eta \leq \eta_l$, an intermittency
or transition regime when $\eta_l < \eta \leq \eta_u$, and unstable regime when $\eta > \eta_u$.

**5. Discussion**
As mentioned above, the natural period of the one-body system at low
displacements is $T_o=2\pi/\omega_o=2\pi(m/K)^{1/2}$ in the absence of friction and viscosity and
$T_n=2\pi/\omega_n=T_o/[1-T_o^2(\eta^2+4m\gamma)/(4\pi m)^2]^{1/2}$ in the presence of friction and viscosity.
Due to $\gamma=1/u_c$ at $u=0$, $T_n$ increases with decreasing $u_c$. Obviously, $T_n$ is longer than
$T_o$ and increases with $\eta$ and $\gamma$, thus indicating that friction and viscosity both lengthen
the natural period of the system.
Based on the marginal analysis of the normalized equation of motion, i.e., Eq.
(11), the plot of $\eta$ versus $1/U_c$ is displayed in Fig. 4 which exhibits the phase portrait
and root structure of the system. Since $\eta$ and $U_c$ are both positive, only the plot of $\eta$
versu $1/U_c$ in the first quadrant is displayed. In Fig. 4, the solid line displays the
function: $D(\eta,1/U_c)=\eta^2-4(1-1/U_c)=0$. Along the line, the solution $\lambda_\pm=-\eta/2$ and thus
$\exp(\lambda t)=\exp(-\eta/2)$. In other word, the roots are equal and real, and, thus, the phase
portrait is a stable inflected node displayed by a solid circle in Fig. 4. Because of $\eta \geq 0$,
we have $1/U_c \leq 1$. As $D(\eta,1/U_c)>0$ or $\eta^2>4(1-1/U_c)$, the roots are both real and
negative. The solution shows no oscillation and thus phase portrait is a stable node
shown by a solid square in Fig. 4. Because of $\eta \geq 0$, we have $1/U_c \leq 1$. As $D(\eta,1/U_c)<0$
or $\eta^2<4(1-1/U_c)$, the roots are complex with a negative real part. This results in
oscillations with exponentially decaying amplitude. The phase portrait is a stable
spiral or a stable focus shown by an open circle in Fig. 4.
Fig. 5 exhibits the time variation in $V/V_{max}$, the phase portrait of $V/V_{max}$ versus
$U/U_{max}$, and Fourier spectrum for four values of $U_c$: (a) for $U_c=0.1$; (b) for $U_c=0.4$; (c)
for $U_c=0.7$; and (d) for $U_c=0.9$ when $\eta=0$. In the first panels, the time variation in
$V/V_{max}$ exhibits cyclic behavior and the amplitude of $V/V_{max}$ decreases and the



predominant period of signal increases with increasing $U_c$. This is consistent with Eq.
(5) in which $T_n$ increases with $U_c$. Although the four phase portraits are almost similar,
yet their size decreases with increasing $U_c$. The second panels exhibit two fixed points:
one at V=0 and U=0 and the second one at larger V and larger V. The slope values at
the first fixed points decrease with increasing $U_c$, thus suggesting that the fixed point
is not an attractor for small $U_c$ and can be an attractor for larger $U_c$. The slope values
at the fixed points for smaller $U_c$ are greater than 1 and thus they cannot be an
attractor. The third panel for each case displays the Fourier spectrum. Fourier spectra
exhibit that in addition to the peak related to the predominant frequency, there are
numerous peaks associated with higher frequencies. This shows nonlinear behavior
caused by nonlinear friction. The frequency related to the first peak decreases with
increasing $U_c$. The amplitude of a peak decreases with increasing $U_c$. The amplitude
of a peak decreases with increasing $\Omega$ for small $U_c$; while it first increases up to the
maximum and then decreases with increasing $\Omega$ for large $U_c$. The amplitude of a peak
becomes very small when $\Omega > 0.25$.

Fig. 6 exhibits the time variation in $V/V_{max}$, the phase portrait of $V/V_{max}$ versus

$U/U_{max}$, and Fourier spectrum for four values of $U_c$: (a) for $U_c$=1.00; (b) for $U_c$=1.01;
(c) for $U_c$=1.15; and (d) for $U_c$=2.0 when $\eta$=0. In the first panels, the time variation in
$V/V_{max}$ exhibits cyclic behavior and the amplitude of $V/V_{max}$ remarkably decreases
with increasing $U_c$ when $U_c$>1. In the second panels, the size of phase portrait
decreases with increasing $U_c$ and there are two fixed points: the first one at V=0 and
U=0 and the second one at larger V and larger V. With comparison to the phase
portrait of $U_c$=1.0, the phase portrait becomes very small when $U_c \geq 1.15$. In contrast
to Fig. 5, the absolute values of slope at the fixed points in Fig. 6 increase with $U_c$.
Hence, the fixed points cannot be an attractor for $U_c \geq 1$. In the third panels, Fourier
spectra exhibit that except for $U_c$=1, there is only one peak and the predominant
frequency increases or the predominant period decreases with increasing $U_c$. This is
consistent with Eq. (5). Results show that nonlinear behavior disappears when $U_c$>1.
In addition, the amplitude of a peak decreases with increasing $U_c$ when $U_c$>1.
Obviously, $U_c$=1 is the critical value of the friction law as displayed in Fig. 4.

Fig. 7 exhibits the time variation in $V/V_{max}$, the phase portrait of $V/V_{max}$ versus

$U/U_{max}$, and Fourier spectrum for four values of $\eta$: (a) for $\eta$=0.20; (b) for $\eta$=0.50; (c)
for $\eta$=0.87; and (d) for $\eta$=0.90 when $U_c$=0.20. In the first panels, the time variation in

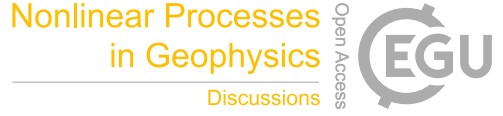

$V/V_{max}$ exhibits cyclic behavior and the amplitude of $V/V_{max}$ decreases with
increasing η. The predominant period of signal only slightly increases with increasing
η, because η changes in a small range. In the second panels, the size of phase portrait
decreases with increasing $U_c$ and there are two fixed points: the first one at V=0 and
U=0 and the second one at larger V and larger V. Since the slope values of fixed
points are clearly all higher than 1, they are not an attractor. In the third panels, the
Fourier spectra exhibit that in addition to the peak related to the predominant
frequency, there are numerous peaks associated with higher Ω. This shows nonlinear
behavior, mainly caused by nonlinear friction, of the model. The highest peak for case
(a) appears at the second frequency. When η<0.9, the amplitude of a peak decreases
with increasing η. The frequencies related to the peaks do not change remarkably,
because η varies in a small range. Except for case (a), the amplitude of a peak
decreases with increasing Ω. The third peak amplitude disappears when η>0.5. The
amplitude of a peak becomes very small when Ω>0.25. Except for $U_c$=0.1, the
frequencies related to the peaks in Fig. 7 are different from and slightly smaller than
those in Fig. 5. Note that when $U_c$<0.55 the simulation results in Fig. 5 are similar to
those in Fig. 6.
Fig. 8 shows the time variation in $V/V_{max}$, the phase portrait of $V/V_{max}$ versus
$U/U_{max}$, and Fourier spectrum for four values of η: (a) for η=0.46; (b) for η=0.47; (c)
for η=0.98; and (d) for η=0.99 when $U_c$=0.55. When η≤0.47, the time variation in
$V/V_{max}$ exhibits cyclic oscillations specified with different main angular frequencies.
When η>0.47 (for example η=0.98 in the figure), in addition to cyclic behavior there
is small intermittent slip with shorter periods. This phenomenon also exists when
$η_l$<η<$η_u$=0.98. There are unstable (or chaotic) slip when η>$η_u$. Hence, the phase
portraits in the second panels display unstable slip at small V and U when
$η_l$<η≤$η_u$=0.98. When η=0.99, the velocity becomes negative infinity at a certain time
and the phase portrait also displays unstable or chaotic slip at small V and U. Since
the slope values of fixed points at large V and U are clearly higher than 1, they are not
an attractor. Due to the appearance of infinity velocity when η=0.99, the Fourier
spectrum is not calculated for η=0.99. The Fourier spectra exhibit that when η<0.47,
in addition to the peak related to the predominant frequency, there are numerous peaks
associated with higher Ω. This shows nonlinear behavior of the model caused by
nonlinear friction. The amplitude of a peak decreases with increasing $U_c$ and the peak



amplitude decreases with increasing $\Omega$. When $\eta=0.98$, the amplitude of the highest
peak is much larger than others. For the first three cases, the amplitude of a peak
becomes very small when $\Omega>0.25$. The frequencies related to the peaks in Fig. 8 are
different from and slightly smaller than those in Fig. 7.

Figs. 9–12 show a variation from stable slip to intermittent slip and then to

unstable or chaotic slip when $\eta$ increases from a smaller value to a larger one for
$U_c=0.6, 0.7, 0.8$, and $0.9$. The values of $\eta_u$ for $U_c=0.20$–$0.95$ with a unit difference of
$0.05$ are given in Table 1. Like Fig. 8, when $\eta\leq\eta_l$, the time variation in $V/V_{max}$
exhibits only cyclic oscillations specified with different frequencies. When $\eta_l<\eta\leq\eta_u$,
there are small intermittent displacements appear in the cyclic oscillations. Hence, the
phase portraits display that unstable slip at small $V$ and $U$ when $\eta_l<\eta\leq\eta_u$. When
$\eta>\eta_u$, the velocity becomes negative infinity at a certain time and the phase portrait
displays unstable slip at small $V$ and $U$. Due to the appearance of infinity velocity, the
Fourier spectrum is not calculated for $\eta>\eta_u$. When $\eta<\eta_l$, in addition to the peak
related to the predominant frequency, there are numerous peaks related to higher $\Omega$.
This shows nonlinear behavior, mainly caused by nonlinear friction, of the model. The
amplitude of a peak decreases with increasing $U_c$ and the amplitude of a peak
decreases with increasing $\Omega$. For the first three cases, the amplitude of a peak
becomes very small when $\Omega>0.25$. Figs. 7–12 show that the frequencies related to the
peaks slightly decrease with increasing $U_c$ and the decreasing rate decreases with
increasing $U_c$. In other word, the frequencies related to the peaks for large $U_c$ are
almost similar. The number of higher peaks and the amplitudes of peaks at higher $\Omega$
both decrease with increasing $\eta$. This indicates that viscosity makes a stronger effect
on higher- frequency waves than on lower ones, and the effect increases with $\eta$.

Fig. 13 exhibits the data points of $\eta_l$ (with a solid square) and that of $\eta_u$ (with a

solid circle) for several values $U_c$. The values of $\eta_l$ and $\eta_u$ for several values of $U_c$
are given in Table 1. The figure exhibits a stable regime when $\eta\leq\eta_l$, an intermittency
(or transition) regime when $\eta_l<\eta\leq\eta_u$, and unstable (or chaotic) regime when $\eta>\eta_u$.
When $U_c<0.55$, there is no $\eta_l$, in other word, unstable slip does not exist. Clearly, $\eta_l$,
$\eta_u$, and their difference $\eta_u-\eta_l$ all decrease with increasing $U_c$. This means that it is
easier to yield unstable slip for larger $U_c$ than for smaller $U_c$. Since smaller $U_c$ is
associated to larger $\gamma$ of decreasing rate of friction force with slip, it is easier to yield



unstable slip from smaller γ than from larger γ.

Huang and Turcotte (1990, 1992) observed intermittent phases in the

displacements based on a two-body model. In other word, some major events are
proceeded by numerous small events. Those small events could be foreshocks. They
also claimed that earthquakes are an example of deterministic chaos. Ryabov and Ito
(2001) also found intermittent phase transitions in a two-dimensional one-body model
with velocity-weakening friction. Their simulations exhibit that intermittent phases
appear before large ruptures. From numerical simulations of earthquake ruptures
using a one-body model with a rate- and state-friction law, Erickson et al. (2008)
found that the system undergoes a Hopf bifurcation to a periodic orbit. This periodic
orbit then undergoes a period doubling cascade into a strange attractor, recognized as
broadband noise in the power spectrum. From numerical simulations of earthquake
ruptures using a two-body model with a rate- and state-friction law, Abe and Kato
(2013) observed various slip patterns, including the periodic recurrence of seismic and
aseismic slip events, and several types of chaotic behavior. The system exhibits
typical period-doubling sequences for some parameter ranges, and attains chaotic
motion. Their results also suggest that the simulated slip behavior is deterministic
chaos andt ime variations of cumulative slip in chaotic slip patterns can be well
approximated by a time-predictable model. In some cases, both seismic and aseismic
slip events occur at a slider, and aseismic slip events complicate the earthquake
recurrence patterns. The present results seem to be comparable with those made by
the previous authors, even though viscosity was not included in their studies. This
suggests that nonlinear friction and viscosity play the first and second roles,
respectively, on the intermittent phases. The intermittent phases could be considered
as foreshocks of the mainshock which is associated with the main rupture. Simulation
results exhibit that foreshocks happen for some mainshcoks and not for others.

**6. Conclusions**

In this work, multistable slip of earthquakes caused by slip-weakening friction

and viscosity is studied based on the normalized equation of motion of a one-degree-
of-freedom spring-slider model in the presence of the two factors. The friction is
caused by thermal pressurization and decays exponentially with displacement. The
major model parameters are the normalized characteristic distance, $U_c$, for friction



and the normalized viscosity coefficient, $\eta$, between the slider and the background
moving plate, which exerts a driving force on the former. Analytic results at small U
suggest that there is a solution regime for $\eta$ and $\gamma$ (=$1/U_c$) to make the slider slip
steadily. Numerical simulations lead to the time variation in $V/V_{max}$, the phase portrait
of $V/V_{max}$ versus $U/U_{max}$, and Fourier spectrum. Results show that the time variation
in $V/V_{max}$, obviously depends on $U_c$ and $\eta$. The effect on the amplitude is stronger
from $\eta$ than from $U_c$. When $U_c>1$, the time variation of $V/V_{max}$ exhibits cyclic
oscillations with a single period and the amplitude of $V/V_{max}$ remarkably decreases
with increasing $U_c$. When $U_c<1$, slip changes from stable motion to intermittency and
then to unstable motion when $\eta$ increases. For a certain $U_c$, the three regimes are
controlled by a lower bound, $\eta_l$, and an upper bound, $\eta_u$, of $\eta$. When $U_c<0.55$, $\eta_u$ is
absent and thus unstable or chaotic slip does not exist. When $U_c\geq0.55$, the plots of $\eta_l$
and $\eta_u$ versus $U_c$ exhibit a stable regime when $\eta\leq\eta_l$, an intermittency (or transition)
regime when $\eta_l<\eta\leq\eta_u$, and unstable (or chaotic) regime when $\eta>\eta_u$. The values of $\eta_l$,
$\eta_u$, and $\eta_u$-$\eta_l$ all decrease with increasing $U_c$, thus suggesting that it is easier to yield
unstable slip for larger $U_c$ than for smaller $U_c$ or larger $\eta$ than for smaller $\eta$. The
phase portraits of $V/V_{max}$ versus $U/U_{max}$ exhibit that there are two fixed points: The
first one at large $V/V_{max}$ and large $U/U_{max}$ is not an attractor for all cases in study;
while the second one at small $V/V_{max}$ and small $U/U_{max}$ can be an attractor for some
values of $U_c$ and $\eta$. When $U_c<1$, the Fourier spectra calculated from simulation
velocity waveforms exhibit several peaks rather than one, thus suggesting the
existence of nonlinear behavior of the system. When $U_c>1$, the related Fourier spectra
show only one peak, thus suggesting linear behavior of the system.

**Acknowledgments**. The study was financially supported by Academia Sinica, the
Ministry of Science and Technology (Grant No.: MOST-105-2116-M-001-007), and
the Central Weather Bureau (Grant No.: MOTC-CWB-106-E-02).

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








Table 1. Values of $\eta_l$, $\eta_u$, and $V_{max}$ for various $U_c$.

| $U_c$ | $\eta_l$ | $\eta_u$ | $V_{max}$ |
|-------|----------|----------|-----------|
| 0.20 | 0.87 | 1.00 | 0.4068 |
| 0.25 | 0.86 | 1.00 | 0.3611 |
| 0.30 | 0.86 | 1.00 | 0.3149 |
| 0.35 | 0.77 | 1.00 | 0.2905 |
| 0.40 | 0.69 | 1.00 | 0.2649 |
| 0.45 | 0.57 | 1.00 | 0.2497 |
| 0.50 | 0.51 | 1.00 | 0.2216 |
| 0.55 | 0.43 | 0.98 | 0.1989 |
| 0.60 | 0.39 | 0.84 | 0.1684 |
| 0.65 | 0.38 | 0.78 | 0.1338 |
| 0.70 | 0.34 | 0.72 | 0.1071 |
| 0.75 | 0.26 | 0.69 | 0.0879 |
| 0.80 | 0.25 | 0.55 | 0.0604 |
| 0.85 | 0.18 | 0.48 | 0.0423 |
| 0.90 | 0.14 | 0.37 | 0.0234 |
| 0.95 | 0.12 | 0.25 | 0.0076 |










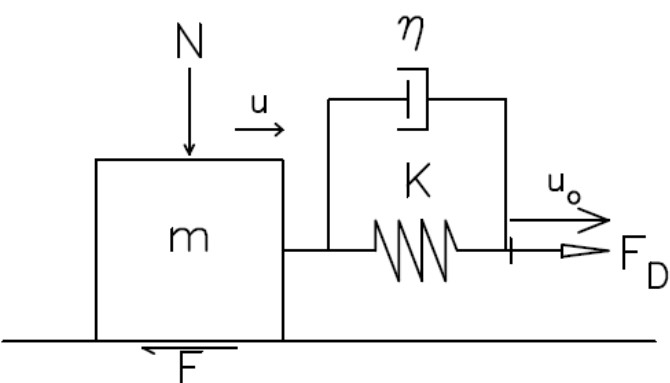

Figure 1. One-body spring-slider model. In the figure, u, K, η, $F_D$, N, and F denote,
respectively, the displacement, the spring constant, the viscosity coefficient, the
driving force, the normal force, and the frictional force.






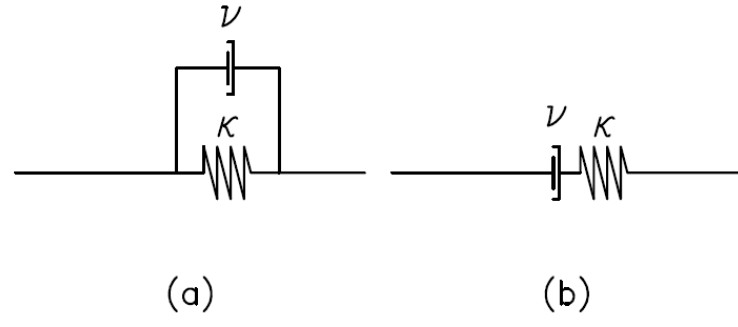


Figure 2. The two types of viscous materials: (a) for the Kelvin–Voigt model and (b)

for the Maxwell model. ($\kappa$=spring constant and $\upsilon$=coefficient of viscosity)









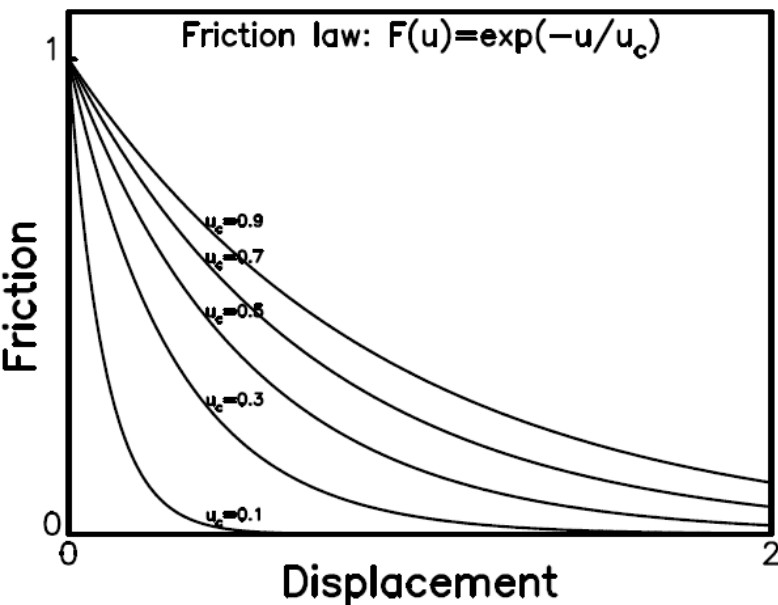


Figure 3. The variations in friction force with displacement for F(u)=exp(-u/$u_c$) when
$u_c$=0.1, 0.3, 0.5, 0.7, and 0.9 m (after Wang, 2016b).








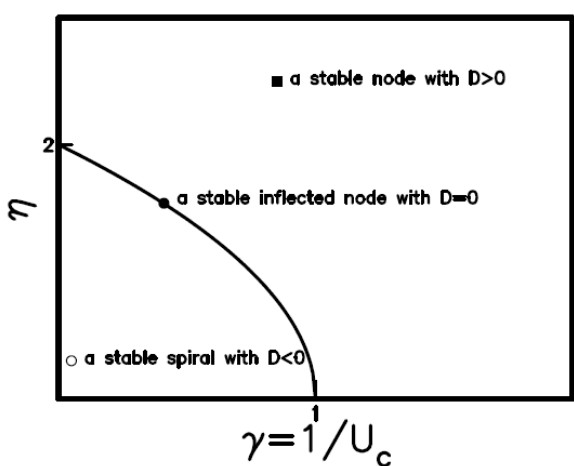

Figure 4. The plot of η versus $1/U_c$ exhibits the phase portrait and root structure of the
system. The solid line displays the function: $D(\eta, 1/U_c) = \eta^2 - 4(1 - 1/U_c) = 0$. The
solid circle, open circle, and solid square represent, respectively, a stable
inflected node with D=0, a stable spiral with D<0, and a stable node with D>0.






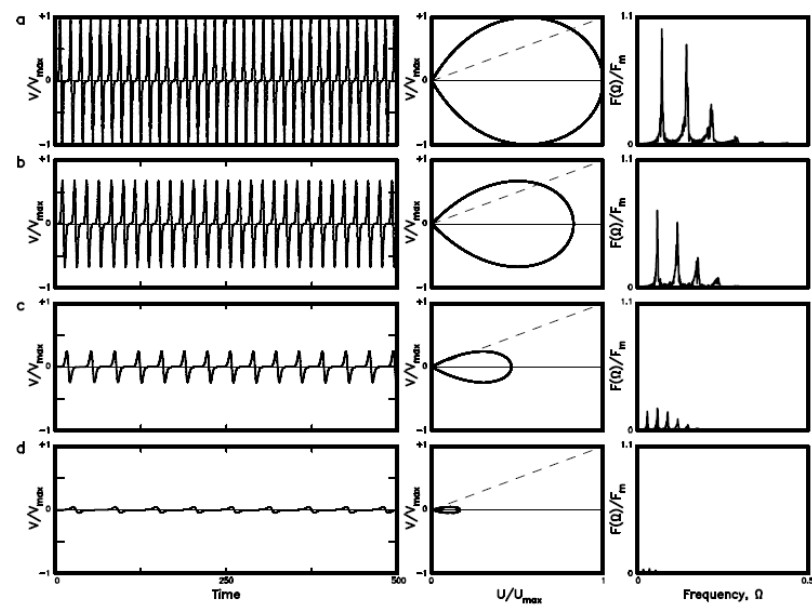

Figure 5. The time variation in $V/V_{max}$, the phase portrait of $V/V_{max}$ versus $U/U_{max}$,
and power spectrum for four values of $U_c$: (a) for $U_c$=0.1; (b) for $U_c$=0.4; (c) for
$U_c$=0.7; and (d) for U =0.9 for the TP law of $F(U)=\exp(-U/U_c)$ when $\eta$=0.



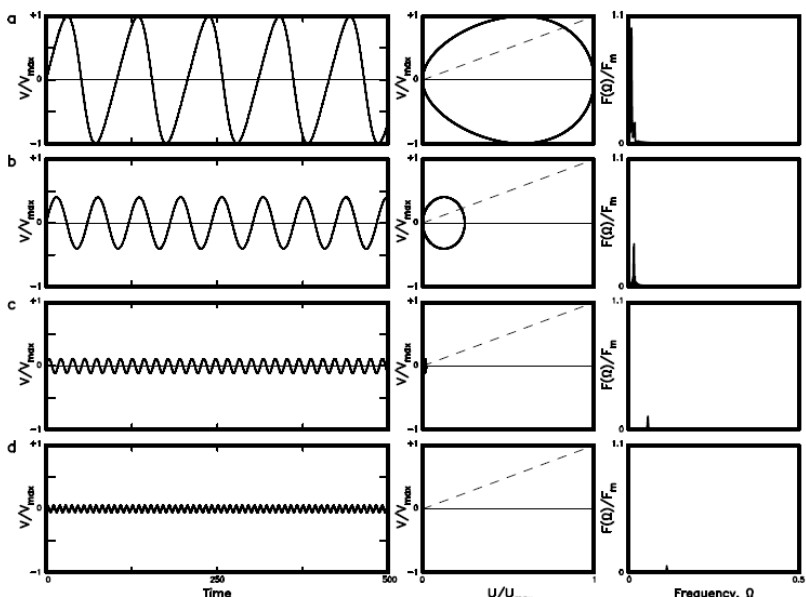

Figure 6. The time variation in $V/V_{max}$, the phase portrait of $V/V_{max}$ versus $U/U_{max}$,
and power spectrum for four values of $U_c$: (a) for $U_c$=1.00; (b) for $U_c$=1.01; (c)
for $U_c$=1.15; and (d) for U =2.00 for the TP law of $F(U)=\exp(-U/U_c)$ when $\eta$=0.






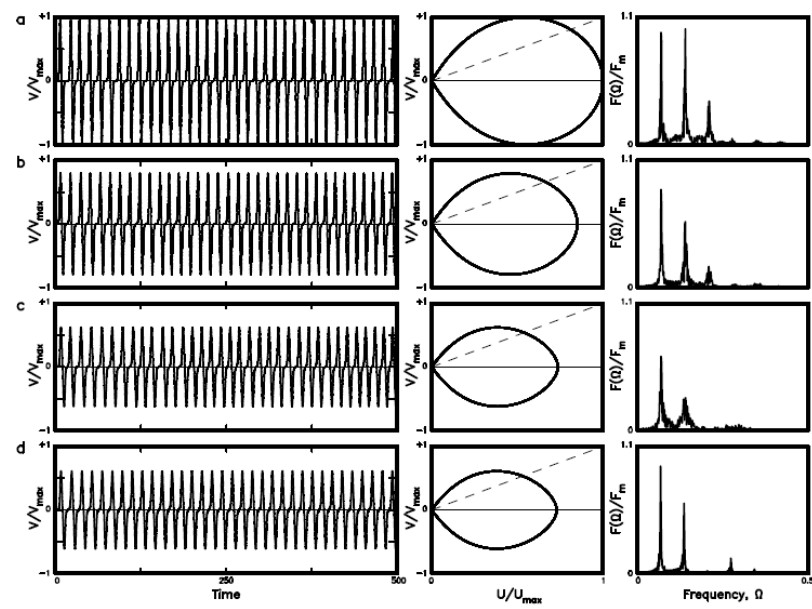

Figure 7. The time variation in $V/V_{max}$, the phase portrait of $V/V_{max}$ versus $U/U_{max}$,
and power spectrum for four values of η: (a) for η=0.20; (b) for η=0.50; (c) for
η=0.87; and (d) for η=0.90 when $U_c$=0.20 for the TP law of $F(U)$=exp($-U/U_c$).







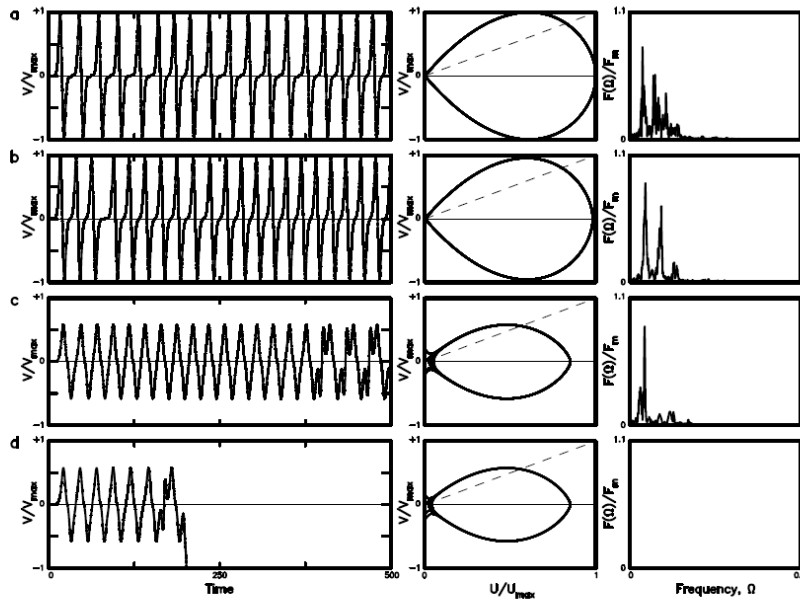


Figure 8. The time variation in $V/V_{max}$, the phase portrait of $V/V_{max}$ versus $U/U_{max}$,
and power spectrum for four values of $\eta$: (a) for $\eta=0.43$; (b) for $\eta=0.47$; (c) for
$\eta=0.98$; and (d) for $\eta=0.99$ when $U_c=055$ for the TP law of $F(U)=\exp(-U/U_c)$.






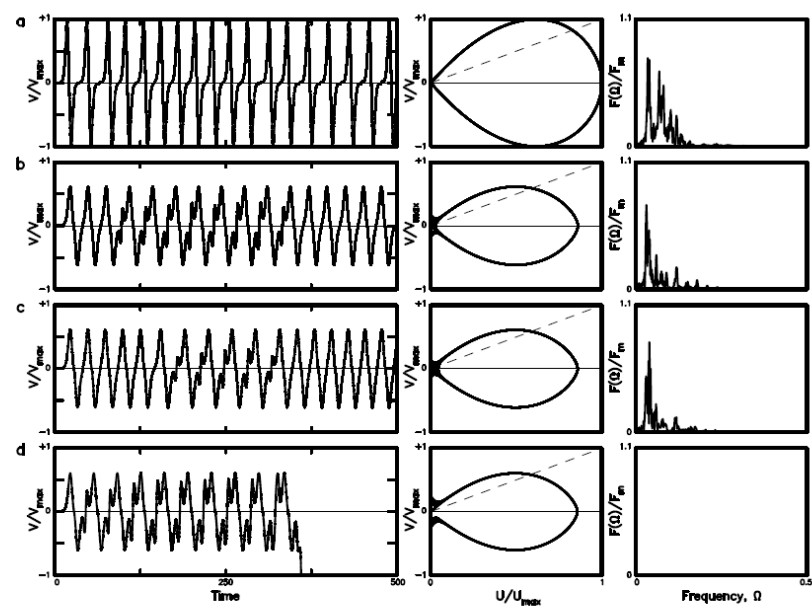

Figure 9. The time variation in $V/V_{max}$, the phase portrait of $V/V_{max}$ versus $U/U_{max}$,
and power spectrum for four values of $\eta$: (a) for $\eta=0.39$; (b) for $\eta=0.83$; (c) for
$\eta=0.84$; and (d) for $\eta=0.85$ when $U_c=0.6$ for the TP law of $F(U)=\exp(-U/U_c)$.









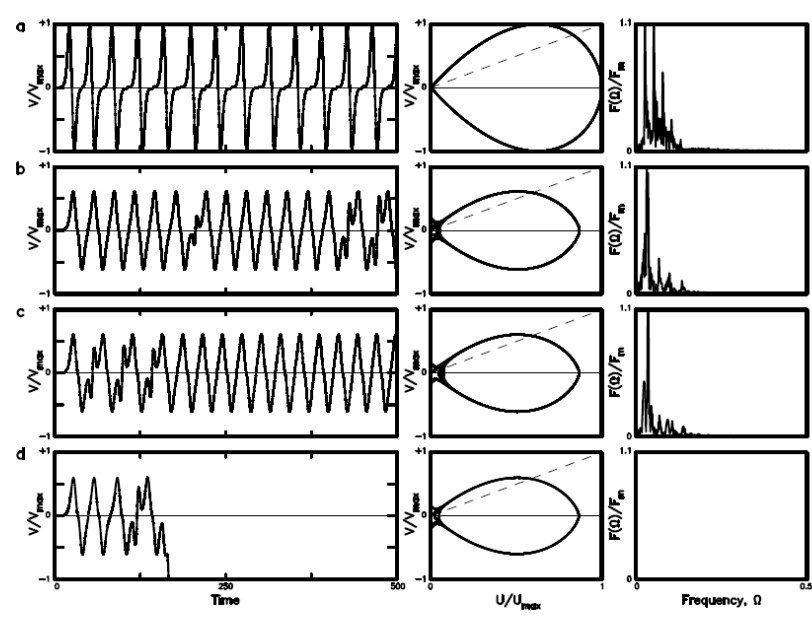

Figure 10. The time variation in $V/V_{max}$, the phase portrait of $V/V_{max}$ versus $U/U_{max}$,
and power spectrum for four values of $\eta$: (a) for $\eta=0.34$; (b) for $\eta=0.71$; (c) for
$\eta=0.72$; and (d) for $\eta=0.73$ when $U_c=0.7$ for the TP law of $F(U)=\exp(-U/U_c)$.








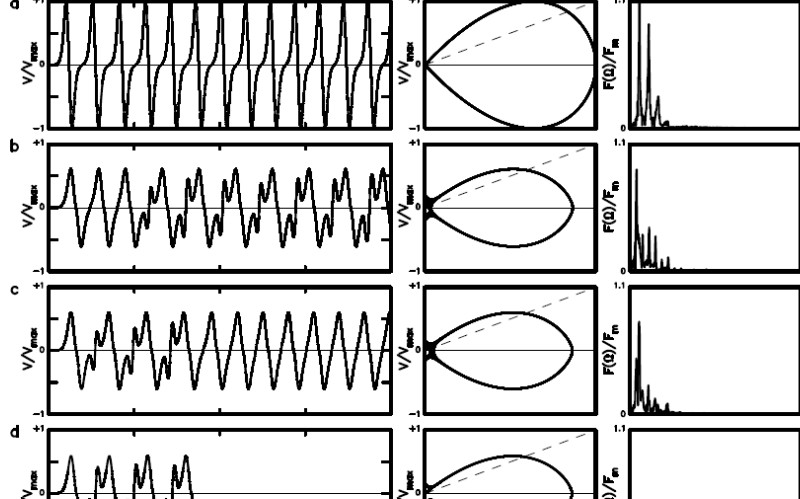

Figure 11. The time variation in V/V$_{max}$, the phase portrait of V/V$_{max}$ versus U/U$_{max}$,
and power spectrum for four values of η: (a) for η=0.25; (b) for η=0.54; (c) for
η=0.55; and (d) for η=0.56 when U$_c$=0.8 for the TP law of F(U)=exp(-U/U$_c$).








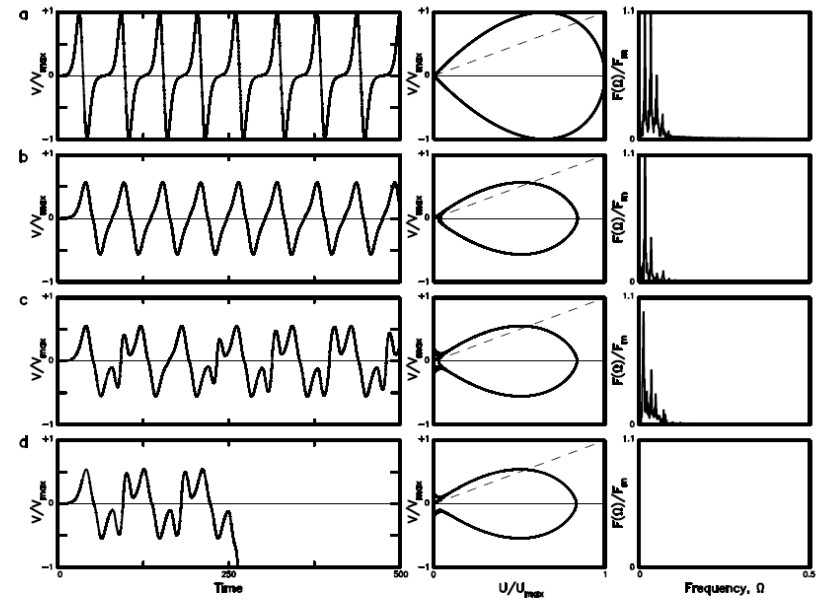

Figure 12. The time variation in V/V$_{max}$, the phase portrait of V/V$_{max}$ versus U/U$_{max}$,
and power spectrum for four values of η: (a) for η=0.14; (b) for η=0.36; (c) for
η=0.37; and (d) for η=0.38 when U$_c$=0.9 for the TP law of F(U)=exp(-U/U$_c$).






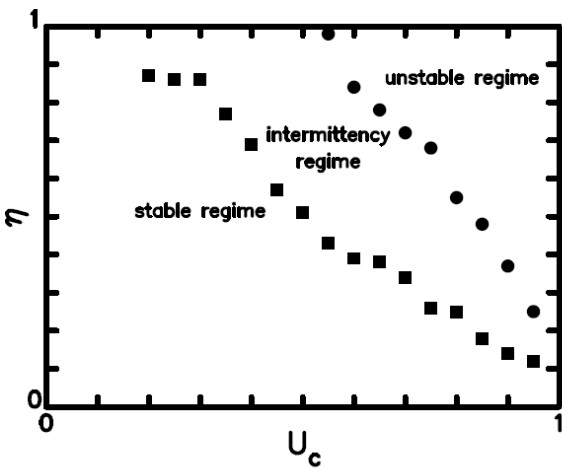

Figure 13. The plot of $\eta_l$ (with a solid square) and $\eta_u$ (with a solid circle) versus $U_c$.