# Peer review of "Multistable Slip of a One-degree-of-freedom Spring-slider"

_Nonlinear Processes in Geophysics, 2017_

## Referee Comment (RC1) · J. G. Spray (Referee) · 1 Jun 2017

[referee-annotated manuscript omitted]

---

## Author Comment (AC1) · 2 Jun 2017

Prof. Spray addressed an important problem about the AUD model. This model is simple and corresponds to a homogeneous simple shear strain at a constant normal stress on a spatial scale of the sheared layer that is broad enough to effectively preclude heat or fluid transfer. Actually, it cannot completely represent the whole behavior of thermal pressurization (TP) and is only an end model of TP model. Unlike the slip-on-a-plane (SOP) model, the AUD model allows varying velocities during faulting and there is a sheared layer, which is comparable with a real fault, even though it is not related to displacement. Meanwhile, Rice (2006) has provided a well-defined equation to describe

the AUD model. This will be easier for me to perform numerical simulations.

This study is just the first step for me to investigate the effect caused by thermal-pressurized friction on earthquake ruptures. This will educate me how to handle and understand the thermal-pressurized effect. My next step will apply the comprehensive TP model proposed by Rice (2006) to one-body and many-body spring-slider models to explore thermal-pressurized effect on earthquake ruptures.

Could you let me know, if you assume that I have not yet completely answered your question? I will try to say something more.

Thanks for your "minor edits" in the PDF file. They are useful for me to improve the manuscript.

---

## Author Comment (AC2) · 19 Jun 2017

General Comments: This is a well-documented and well-written article, however much of it seems to dwell on and define standard mathematical concepts (like deterministic chaos, fixed points and attractors) and seems not appropriate to include in a research publication. Furthermore, a parameter study (as done in this paper) finds different solution style regimes and a linearized stability analysis (which would determine critical parameter values for which bifurcations occur) seems warranted. [Answer] Thanks for valuable comments. This work is focused on basic studies of slip of a one-body model in the presence of thermal pressurized slip-weakening friction and viscosity. The main

purpose is the identification the solution regimes in terms of model parameters through theoretical marginal analyses and numerical simulations. Hence, it is necessary to use much mathematics. The identification of solution regimes will be useful for the understanding nonlinear earthquake ruptures. The study concerning nonlinear rupture processes of earthquakes will be my next step.

Specific comments: Please define the word "multistable." It is not a commonly used term. [Answer] It is a mathematical term representing a (nonlinear) system having multiple states of stability or instability. In this work, I study the solution regimes: stable, intermittent, and unstable. Hence, the word "multistable" is taken into account.

Lines 157–159 need to specify when this is true. As I recall, this situation occurs under a constant tensile stress. [Answer] You are right. This situation occurs under a constant tensile stress. The statements are re-written as "Under a constant tensile stress, the strain will increase, without a upper limit, with time for the Maxwell model; while the strain will increases, with a upper limit, with time for the Kelvin-Voigt model."

Line 161-162: if viscosity is taken to be constant throughout the work, then paragraph 129-145 is inappropriate, or at least should be cut down or moved to a section on future work. [Answer] The statements shown in Lines 161–162 are used to stress that the main factor in influencing viscosity is temperature. Temperature actually varies during an earthquake rupture. Hence, it should be better to consider varying viscosity coefficients due to the temperature change during a rupture process in a numerical simulation. However, in this study a main attempt is made to search for the solution regimes which are in terms of the main parameter of friction and that of viscosity. Hence, the value of viscosity coefficient is set to be a constant in a numerical simulation. Actually, the statement written in Lines 161–162 is not written clearly. The statement should be re-written as "To simplify the problem, only a constant viscosity coefficient is considered in a numerical simulation as given below."

Example of too much attention paid to well-known results/definitions, for example,

NPGD
10.5194/npg-2017-17
2017

equation (9) is a linear, second order homogeneous ODE, so why include the discussion of its solution if this is a standard, textbook exercise? [Answer] The statements about Equation (9) are the theoretical marginal analyses of the problem. Just like your comment, they can be found in a standard textbook because only the solution types of an ODE under different values of parameters are described. However, to retain completeness of the work about the theoretical marginal analyses and to help some readers who are not so familiar with ODE. I wish to keep the statements. Of course, the statements can be largely reduced in case that the reviewer requests.

Line 317-318: this phrase "the velocity becomes negative infinity at a certain time" appears several times throughout the manuscript, with little explanation as to why. Is the problem ill-posed in this parameter regime? [Answer] This statement is just used to indicate the unstable rupture appearing in the unstable regime. It is OK to say that the problem becomes ill-posed in this solution regime. The statements "... the velocity becomes negative infinity at a certain time and the phase portrait also displays unstable or chaotic slip at small V and U" are not clear enough, and thus will be re-written as "... the velocity becomes an abnormal large negative value at a certain time and the phase portrait also displays unstable or chaotic slip at small V and U. This exhibits unstable slip of the system. In other word, the problem becomes ill-posed in this parameter regime" in the revised manuscript.

Technical corrections: there are many typos throughout the manuscript. [Answer] Reviewer #1 has substantially pointed out the typo errors in her/his comments.

Please also note the supplement to this comment:
http://www.nonlin-processes-geophys-discuss.net/npg-2017-17/npg-2017-17-AC2-supplement.pdf